# The Potential of Acknowledging the Unknown: Single Positive Multi-label Learning in Medical Image Processing

**Helen Schneider**[*1]                    HELEN.SCHNEIDER@IAIS.FRAUNHOFER.DE
[1] *Fraunhofer Institute for Intelligent Analysis and Information Systems IAIS*

**Priya Priya**[*1,2]
[2] *University of Bonn*                    PRIYA.PRIYA@IAIS.FRAUNHOFER.DE
**Aditya Parikh**[1]                        ADITYA.PARIKH@IAIS.FRAUNHOFER.DE
**Christian Bauckhage** [1,2]         CHRISTIAN.BAUCKHAGE@IAIS.FRAUNHOFER.DE
**Rafet Sifa** [1,2]                        RAFET.SIFA@IAIS.FRAUNHOFER.DE

**Editors:** Under Review for MIDL 2024

## Abstract

In the wake of high complexity and resource constraints, the manual annotation of the medical images involves a severe trade-off between the annotated labels per sample and the dataset size, besides inducing a high count of false-negative labels due to the underlying intricacies in recognizing all potential pathologies or conditions. Single Positive Multi-Label (SPML) learning aims to address the case of label noise by using only a single positive label per sample for the training, considering the remaining labels as uncertain. As SPML is yet to be fully explored in the realm of medical imaging, therefore, this work focuses on investigating the state-of-the-art SPML loss functions subsuming the Generalized Assume Negative (G-AN) and Entropy Maximization (EM) losses on different training sample counts. Additionally, we investigate the influence of the asymmetric pseudo-labeling with EM loss in minimizing the effect of label uncertainty induced by SPML learning.

**Keywords:** Single positive multi-label training, Medical imaging, Noise-robust training, Entropy-maximization loss, Asymmetric pseudo-labeling, Generalized assume negative loss

## 1. Introduction

Manual annotation of multi-label datasets is a costly and time-consuming task, often resulting in a high number of false negative labels due to oversight of presented labels (Cole et al., 2021). Additionally, there is often a trade-off between the number of considered labels and annotated samples due to time and cost constraints (Deng et al., 2014). This also applies in particular to multi-label medical imaging datasets. Single positive multi-label (SPML) training addresses the most severe version of these problems, where only one positive label is observed for each sample, and the other labels are unknown (neither negative nor positive) (Cole et al., 2021). While comparable performance with significantly lower annotation effort has been achieved for natural images through SPML training, SPML training for medical imaging has not yet been sufficiently explored, even though it represents a domain where particularly costly annotation of data is required, partly due to the frequently required

---

[*] Contributed equally

expertise of the annotators (Durand et al., 2019)(Nowak et al., 2023). This work attempts to bridge this gap by analyzing state-of-the-art SPML training approaches for small-sized medical imaging datasets.

We therefore investigate the SPML loss functions on medical images against the standard baselines of multi-label Binary Cross Entropy (BCE) and Assume Negative (AN) loss functions, where each unknown label is mapped to the more likely negative label (Cole et al., 2021). Additionally, we analyze the influence of more robust SPML loss functions like the Generalized Assume Negative (G-AN) and the Entropy Maximization loss (EM), both based on the Assume Negative Approach (Schneider et al., 2023). Additionally, we evaluate the potential of asymmetric pseudo-labeling (APL) modules with the EM loss and define future work to further improve SPML training for medical imaging (Zhou et al., 2022).

## 2. Methods and Experiments

For our experiments, we use the publicly available CheXpert dataset (Irvin et al., 2019). The train set consists of 224,316 chest radiographs, labeled automatically using corresponding text reports for 14 diagnostic categories as positive, negative, or uncertain. The validation and test sets are annotated manually by the experts and include 202 and 500 images respectively. We focus on classifying five clinically relevant diseases: Atelectasis, Cardiomegaly, Consolidation, Edema, and Pleural Effusion. We consider frontal view samples with at least one positive label and process uncertain labels as negative. We select 5K, 10K, 25K, and 40K training images to analyse the performance with respect to different sample counts.

To evaluate SPML training performance against the conventional multi-label baseline, we use the BCE loss and train the model with all available labels. For SPML, we randomly consider one positive label per sample and map the remaining labels to negative. We consider AN loss as an SPML baseline where the model is trained using BCE loss with a single positive label per sample. This might yield good results since a large portion of labels in multi-label datasets are negative (Cole et al., 2021). Additionally, we evaluate the EM loss for SPML (Zhou et al., 2022) to examine its influence on medical images. We then apply their asymmetric pseudo-labeling method with EM, indicated by EM+APL. Besides, we investigate the G-AN loss function (Schneider et al., 2023) which aims to enforce robustness against label noise. The goal of applying the EM and G-AN losses is to reduce the influence of incorrect negative labels in SPML and boosting the performance up to the level of baseline BCE with all the available labels. Please refer to (Zhou et al., 2022) and (Schneider et al., 2023) for more detailed information about the EM+APL and G-AN loss functions respectively.

All experiments are based on the DenseNet-121 architecture with ImageNet pre-trained weights. Images are preprocessed and rescaled to $224 \times 224$, normalized by ImageNet mean and standard deviation. We use a batch size of 128 samples and the AdamW optimizer with the fixed learning rate of $1 \times 10^{-3}$ and seed for reproducibility. The down-weight strengths for EM loss and EM+APL loss are fixed as 0.1 and 0.9 respectively. The negative sample proportion for unannotated labels for EM+APL is set to 0.9. We keep the test and validation set unaltered for evaluation. Evaluation is conducted by selecting the best model based on the validation AUROC score and further evaluated on the test set.

## 3. Results and Conclusion

Table 1 presents the AUROC scores of the above described experiments. As expected, the increase in the sample count reflects an improvement in the AUROC scores. The second anticipated outcome is that EM training improves the performance compared to the AN baseline in three cases except for 25K samples where it is almost similar. This implies that EM is a suitable SPML loss for real life datasets.

When EM is combined with pseudo labels (EM+APL), it improves the scores compared to the EM loss for three of four simulations, indicating that the APL module can potentially benefit the SPML training of small medical datasets. Despite the pseudo label module, the EM loss does not achieve comparable performance to the superior G-AN loss. In three out of four cases, the best AUROC score is achieved with G-AN training. This illustrates the leading performance of G-AN loss for small medical imaging datasets. Future work will be to combine the superior G-AN loss function with an APL module to further investigate the performance improvement in SPML training.

Table 1: AUROC scores on the CheXpert test set for different loss functions when the model is trained with the indicated sample counts. BCE indicates the scores when the model is trained with all the available labels (positive and negative). The remaining cases exhibit the SPML loss function when the model is trained with only one single positive label per sample and other labels are unobserved.

| Samples | BCE | AN | EM | EM+APL | G-AN |
|---------|-----|-----|-----|--------|------|
| 5K | 71.7 | 60.8 | 61.9 | **64.6** | 62.8 |
| 10K | 73.9 | 60.7 | 63.7 | 63.7 | **69.9** |
| 25K | 77.0 | 62.6 | 62.5 | 66.9 | **70.4** |
| 40K | 78.4 | 66.1 | 69.0 | 69.3 | **72.2** |

To conclude our work, we have explored novel approaches for addressing the challenges posed by SPML in medical imaging. Our experiments have demonstrated the efficacy of the asymmetric pseudo-labelling technique for EM loss training for small medical imaging datasets, and the superiority of the G-AN loss, even without the pseudo-label technique. It lays the groundwork for future research in SPML and serves as a robust baseline for further investigations into weakly supervised learning paradigms. Drawing on the success of APL from our study, we suggest exploring and developing its integration with G-AN and additional SPML loss functions to further enhance the performance.

## Acknowledgments

This research has been funded by the Federal Ministry of Education and Research of Germany and the state of North-Rhine Westphalia as part of the Lamarr-Institute for Machine Learning and Artificial Intelligence, LAMARR22B.

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
