# OpenReview forum: "The Potential of Acknowledging the Unknown: Single Positive Multi-label Learning in Medical Image Processing"
_MIDL.io/2024/Short_Papers — MIDL 2024 Short Papers_

### Official Review · Reviewer_GRLK · 2024-04-18

**Confidence:** 4
**Final Rating:** 3.5

**Review:**

The paper addresses a multi-label learning scenario where only one of the labels is considered as certain and the others as unknown. They investigate different loss functions in this setting for a chest x-ray dataset, a loss function based on GANs has the highest AUC.

Strengths:
-	Interesting learning scenario
-	Despite the short length and multiple topics addressed, it is overall clear what is investigated
-	Experiment with increasing sample size

Weaknesses:
-	It seems that the performance could crucially depend on correlations between the labels, and which label is randomly picked by SPML.
-	In general reporting AUC with some kind of variability measure would have been appropriate
-	“significantly improves the scores”  no significance test is described
-	It is difficult to appreciate the differences between the loss functions given the space available, and the presented results


Minor comments:
-	Typo in the title
-	Some details of references are missing

I think the paper is a good fit to the conference so my I think it could be accepted if there is space, but there are also some weaknesses due to which I think other papers might be prioritized more, therefore my slightly conservative score.

---

### Decision · Program_Chairs · 2024-04-26

Accept